

# The longest homogeneous series of grape harvest dates, Beaune 1354-2018, and its significance for the understanding of past and present climate

Thomas Labbé,[1,2] Christian Pfister,[3] Stefan Brönnimann,[3,4] Daniel Rousseau,[5] Jörg Franke,[3,4] Benjamin Bois[6,7]

[1]Leibniz Institute for the History and Culture of Eastern Europe (GWZO), University of Leipzig, Specks Hof, Reichsstraße 4-6, 04109 Leipzig, Germany
[2]Maison des Sciences de l'Homme de Dijon, USR 3516 CNRS, University of Burgundy, 6 Esplanade Erasme, BP 26611, 21066 Dijon cedex, France
[3]Oeschger Centre for Climate Change Research, University of Bern, Hochschustrasse 4, 3012 Bern, Switzerland
[4]Institute of Geography, University of Bern, Hallerstrasse 12, 3012 Bern, Switzerland
[5]Conseil Supérieur de la Météorologie, 73 avenue de Paris, 94160 Saint-Mandé, France
[6]Biogéosciences UMR 6282 CNRS, University of Burgundy, 6 boulevard Gabriel, 21000 Dijon, France
[7]Institut Universitaire de la Vigne et du Vin, University of Burgundy, 1 rue Claude Ladrey, 21000 Dijon, France

*Correspondence to:* thomas.labbe@u-bourgogne.fr

**Abstract**. Records of grape harvest dates (GHD) are the oldest and the longest continuous phenological data in Europe. However, many available series including the well-known (Dijon) Burgundy series are error prone, because scholars so far uncritically drew the data from nineteenth century publications instead of going back to the archives. The GHD from the famous vine region of Beaune (Burgundy) were entirely drawn from the archives, critically cross checked with narrative evidence and calibrated with the long Paris temperature series comprising the 360 years from 1659 to 2018. The 664-year-long Beaune series from 1354 to 2018 is also significantly correlated with tree-ring and documentary proxy evidence as well as with the Central European temperature series (from 1500). The series is clearly subdivided into two parts. From 1354 to 1987 grapes were on average picked from 28 September on, whereby during the last 31 year long period of rapid Global Warming from 1988 to 2018 harvests began 13 days earlier. In the Paris temperature measurements since 1659, April-to-July temperature reached the highest value ever in 2018. The 33 extremely warm events comprising the 5% percentile bracket of GHD are unevenly distributed over time. 21 of them occurred between 1393 and 1719, whereby this is the case for just five years between 1720 and 2002. Since the hot summer 2003 8 out of 16 spring-summer periods were outstanding according the statistic of the last 664 years, no less than 5 among them within the last 8 years. In sum, the 664-year-long Beaune GHD





series demonstrates that outstanding hot and dry years in the past were outliers, whereby they became the norm since the transition to rapid Global Warming in 1988.

## 1. Introduction


Records of grape harvest dates (GHD) provide the longest continuous series of phenological data in Europe and have been repeatedly used for estimating spring-summer temperatures (Chuine et al. 2004; Guiot et al. 2005; Menzel 2005; Le Roy Ladurie et al. 2006; Krieger et al. 2011; Garcia de Cortázar-Atauri et al. 2010; Daux et al. 2012; Rousseau 2015 ). In the literature, the GHD series (1385-1905) from the town of Dijon situated in Burgundy (France, see Fig. 1) is the longest

available homogeneous series. Going back to the late 14[th] century, it constitutes the backbone of all the reconstructions of "Burgundian" GHD series. The most widely quoted article published by Chuine et al. (2004) in "Nature" provides a "Burgundian" GHD dataset (1370-2003) in which Dijon GHD constitute the reference series. Actually, from 1426 to 1700 no other Burgundian GHD than those corresponding to the territory of the city of Dijon are available for scientific purposes. However, two biases affect the reliability of this dataset. First, scholars have until now uncritically drawn on the data from

nineteenth century publications. The original data of the Dijon series have been recently revisited directly in the local archives (Labbé, Gaveau 2011) and this reassessment makes obvious that the formerly published "Dijon series" is thoroughly unreliable, due to compilation misuses. Secondly, the available Burgundian GHD dataset is not homogeneous. Due to a lack of information concerning Dijon where the vineyard has disappeared from the 19[th] century because of the industrialization, the series is complemented for the 18-20[th] centuries by a mix of data taken from different locations

disseminated throughout the vineyard region of Burgundy.

An entirely unedited series based on Manuscript material discovered in the archives of the town of Beaune situated 45 km south of Dijon has then been recently collected for the period 1371-2010 (Labbé, Gaveau 2013). Unlike Dijon, Beaune is still surrounded with vineyards situated on altitudes between 220 and 300 metres. Since the end of the Middle Ages the territory of the city is dominated by the culture of grapes. The region produces labelled wines certified by the "Appellation d'Origine Contôlée" system. In this paper, we have extended the series back to 1354 and updated it to 2018. It is validated

using the long Paris temperature series that goes back to 1658 (Rousseau 2009, Rousseau 2013, updated to 2018) and used to assess April-to-July temperatures from 1354 to 2018. All datasets are available on the "Euro-Climhist" data-plateform: http://www.euroclimhist.unibe.ch/en/ (Federal Office of Meteorology and Climatology MeteoSwiss / Oeschger Centre for Climate Change Research / University of Bern).



The study is organized as follows. The first section reviews the former "Burgundian" GHD series highlighting its inadequacies. The second section presents the generation of the new Beaune series. Section three outlines the methodological steps used for reconstructing April-to-July temperatures from 1354 to 2018 and for checking the reliability of the data. In section four the Beaune series is presented and compared to other GHD series, other documentary data series as well as to tree-ring evidence. This section also provides the reconstruction of April-to-July temperatures from 1354 to 2018 and a

detailed analysis of two extremely early years. In the last section we summarizes the main conclusions that can be drawn from the study, in particular in view of comparing isolated outliers in the remote past with the increasing frequency of such events in the last three decades of rapid Global Warming.

## 2. Data

### 2.1 The inadequacies of the "Dijon/Burgundian" GHD series

Etienne Noirot, land surveyor in Dijon, is one the first scientist**s** who showed some interest in long GHD time-series. In 1836, he gathered a Dijon GHD series from 1385 to his time, mainly to demonstrate that climate did not change significantly since five hundred years (Noirot 1836). In the course of the cold fluctuation leading to the glacier maximum around 1850 (Nussbaumer 2018, 93) GDH's became important for scientists as indicators of past climate (Labbé, Gaveau 2013b). In this context Jules Lavalle (1855) re-published the same dataset for the period 1366-1842. His series forms the backbone of all

further publications using the Dijon evidence. In the early 1880's Alfred Angot, Director of the Paris Meteorological Research Office, instructed the meteorological commissions of the departments to extract grape harvest dates from documentary sources and assembled a compilation of 606 GHD series from France, Switzerland and Germany. This publication is a milestone in the field. The Dijon series, extracted from Jules Lavalle work and continued up to 1879, constitutes the longest series of this dataset (Angot 1885). The French historian Emmanuel Le Roy Ladurie, writing in the

second half of the twentieth century valued the Angot data for his synthesis on climate changes since AD 1000 (Le Roy Ladurie 1971). However, none of his followers ever felt the need to critically assess the original data of the Dijon series. Subsequently, three main sources of inhomogeneity have to be addressed concerning available Burgundian GHD.

### 2.2 Flawed data

First, the "Dijon series" is riddled with printing, typing and copying errors. The investigation in the original archives of the

City Council in Dijon shows132 differences with the Angot series for the period 1385 to 1879 (Labbé, Gaveau, 2011). The mismatches reach 5-10 days for 17 years, 10-20 days in 9 occasions, and more than 20 days in 1448, 1522, 1523, 1540, 1659, 1660 and 1842. A serious mismatch concerns the year 1540. According to the supplementary material on internet the



GHD in this year is 4 October (day-of-year (DOY) 278) (Chuine et al., 2004) whereas the correct value found in the archive
is 3 September (DOY 247). This flaw is the main reason why the outstanding extreme spring-summer temperature of this
year (Wetter and Pfister 2013, Wetter et al. 2014) was overlooked in the "Nature" article by Chuine et al. (2004). In
particular, Angot's series is especially not reliable prior to 1420, in the first half of the sixteenth century, around 1840, and
for several isolated years (Fig. 2).

**2.3 Lack of homogeneity**

The vineyards around Dijon, were built over from the early nineteenth century. In 1906 the City Council did not set official
ban date anymore. For the nineteenth and twentieth century the Dijon series was thus complemented with data from the
southern part of the Burgundian vine growing area without, however, taking into account the resulting differences in mean
grape ripening. It needs to be known that Dijon is situated at the northernmost point of the Burgundian wine region (Fig. 1)
which involves a delay in the mean date of grape harvest compared to other locations along the latitudinal (North-South)
orientation of the Burgundian wine area. Grapes in Dijon between 1600 and 1800 were picked on average 5 days after those
in the town of Beaune (Fig. 3). However the "Burgundian series" used by Chuine et al. (2004), indiscriminately combines
the pre-1800 Dijon series with the 19[th] and 20[th] century evidence from the southern part of the Burgundian vine area. The
resulting temperature reconstruction is therefore inhomogeneous.

**2.4 Assembling a 664 year long GHD series for Beaune**

To correct for the above mentioned inadequacies we constructed a new 664-year long almost homogenous and nearly
uninterrupted series of GHD focussing the archival study on the Beaune vine growing area (available on the "Euro-Climhist"
data-plateform). Thereby we used different kind of archival information.
**Wage payment data (1354-1506)**: Late medieval accounts can be used as a source for day-specific information about the
actual start and end of agricultural work. (Brázdil and Kotyza 1999, Wetter and Pfister 2011, Wetter and Pfister 2013, Pribyl
et al. 2012).
In the case of Beaune, data on daily wage payments made to day-labourers for picking grapes are available from 1354 to
1506. The oldest accounts were kept for a c.18 hectares domain owned by the dukes of Burgundy. The start dates of the
harvest (1354-1426) in these estates have already been published (Guerreau 1995). Nevertheless, the most detailed and
numerous series of accounts refers to the c.10 hectares domain of the church chapter of Notre-Dame in Beaune, whereby we
could find almost without any lacunae unedited GHD from 1371 to 1506. The parcels documented in these accounts are
continuously planted with vines since the late Middle Ages. Thus, these GHD are related to the maturity of grapes in
domains such as *Corton Clos-du-Roi*, *Beaune-Sanvignes*, *Beaune-Tuvillains*, *Beaune-Bressandes* among others, which



produce today first class red wines designated as *Grand cru*, *Premier cru* or *Beaune-village*. The "Beaune" wine produced in these domains had already a reputation for quality since the 13th century (Dion 1959). The chapter of Notre-Dame sold for example barrels of wine to the merchants of the king of France and to other key persons. Undoubtedly, these domains

already produced first class wines. It is however not known which grape varieties were grown in these estates. Fine red and "clairet" (almost rosé) wines massively produced during the Middle Ages, were made with varieties of Pinot noir, whereby Gamay was rated second class. In 1395 the duke of Burgundy even prohibited the cultivation of Gamay vines around the cities of Dijon, Beaune and Chalon-sur-Saône. But in reality his direction never became effective (Dion 1959). In any case, the documentation never refers to the varieties cultivated in the domains.

Nonetheless, these data are very reliable and even refine in which parcel the grapes were picked on a specific day and how many male and female labourers were at work.

 **Meetings of the Notre Dame Church chapter (1507-1699):** In 1506 both aforementioned estates were leased out to tenants, so that precise information on their cultivation are not kept anymore in the accounting documentation. At the same time, the deliberations of the city council containing the setting of the vintage ban are fragmentary prior to 1700. The books

of deliberations of the Notre-Dame Church chapter of Beaune offer an acceptable substitute. This documentation provides the date of the last meeting before the vacation of the chapter who had to organize the harvest. As church chapter meetings took regularly place twice a week, three days were added to the date of the last meeting to assess the harvest date. Since 1583, the books of deliberation indicate at what time the members of the chapter had to organise the food supply needed to feed the day labourers during the harvest. With regard to getting perishable food such as cheese and meat, it was assumed

that the harvest began eight days later.

**Deliberations of the city council (1700 to 1965):** From 1700 onwards the books of deliberation of the city council are continuously preserved and the opening date of the harvest can be easily drawn. However, setting the vintage ban is a social outcome and does not depend only of the observation of the full maturity of grapes. It resulted from a consensus between vine-growers and the local town administration. In practice, experts inspected the maturity of the grapes and proposed a ban

date to the city council. The city council then took a decision, considering also the availability of day labourers coming from outside for harvesting as well as eventual military threats and plague outbreaks, as Garnier and co-authors (2011) have shown for Besançon (eastern France). In the case of Beaune, the main bias that potentially affects the date of setting the ban on the long run is nonetheless different. After the French Revolution actually, the re-organization of the territory into administrative districts called "arrondissements" induced a change in the decision-making process. The dates of the ban now

resulted from a consensus among the mayors of all villages contained in the Beaune "arrondissements". Thought Beaune continues to play the major role in the decision process, the lifting of the harvest ban subsequently concerned a larger area than before extending more to the south. Furthermore, the outbreak of the Phylloxera disease after 1870 induced a new



organization of the vine sector in Burgundy in which local institutions further lost weight in the setting-up of the harvest on their territory. From the second half of the 20th century up to 2007 prefects promulgated a uniform harvest ban which did not any more consider local conditions. Moreover, to avoid dealing with requests for harvesting prior to the official date the prefectoral ban tended to be artificially early. Therefore, another kind of information was used after 1965.

**Newspaper reports (1966 to 2018):** From 1966 articles in the local newspaper "Le Bien Public" are used. Every year an article reports the date when the vine-growers start the harvest in the town vineyard. These dates are closer to the reality than the official ones set by prefectoral decrees. Between 1980 and 2007 the Pearson correlation coefficient r between Beaune "newspaper" GHD and the mean April-to-July (AMJJ) temperatures of Paris is stronger (r = -0.833) than with the official administrative GHD (r = -0.789). In 2007, the official opening of GHD was set on 13th of August whereas wine-growers actually began the harvest in the territory of Beaune on the 1st of September according to the newspapers.

### 2.5 Interpolations

Despite an exhaustive investigation in the local archives, 61 dates for Beaune are still missing before 1645. To fill in these lacunae, when possible we used the evidence of the corrected Dijon series (Labbé, Gaveau 2011), taking into account the mean difference of days between the two raw series.

The dates from 1358 to 1364 have been taken from the Torino GHD series (Rotelli 1973), taking into account the mean differences of days between the two series.

Certain dates are affected by regional political and military biases and need to be interpolated as well with an extra-regional series, e.g. the corrected Swiss series (Wetter and Pfister 2013). The political situation has been particularly critical in the region of Beaune and Dijon in the context of the Thirty Years war (1618-1648). As in the nearby city of Besançon documentary records are sketchy in the second quarter of the 17th century because troop movements often prevented wine-growers to properly organize the harvest (Garnier et al. 2011). In Beaune, the archives do not provide any data from 1631 to 1638 which constitutes the longest undocumented period of the series. We interpolated these data from the Dijon series, but the dates for 1636 (4 September) and 1637 (3 September) still turned out to be artificially early in comparison with the series from the Swiss Mittelland, respectively 5 October and 1 October (Wetter and Pfister 2013). In 1636, the city council deliberations of Beaune inform us that the region was actually threatened by both enemy troops and by an outbreak of plague, which certainly disorganised the harvest process. In 1636 and in 1637, we have then interpolated the dates with the Swiss series.



### 2.6 Homogenisation

A comparison of Beaune GHD times series with the GHD of nearby regions of Switzerland (Wetter, Pfister 2013) and of Salins (Angot 1885), makes obvious that the Beaune GHD are artificially early before c. 1718 (Fig. 4). In the period 1599-1875 for which we can compare the three time series without lacunae, the stability of the mean GHD is stronger in Switzerland and in Salins, whereas in Beaune the average GHD occurred 7 days earlier before 1718 (21 September) than after this date (28 September) (Table 1).

In the perspective of reconstructing past spring-summer temperatures, this bias must be taken into account. To homogenise the Beaune GHD series we have then added 7 days to raw data prior to 1718. The Pearson correlation with April-to-July mean temperature in Paris for the period 1658-2018 is stronger with the homogenised Beaune GHD time series (r = 0.76) than with the non-homogenised one (r = 0.74).

Anthropogenic changes in wine-making are the most likely explanation. The early 18[th] century is a turning point in wine history. The production and the commercialization of wines shifted to a new model distinguishing between ordinary and fine wines and focussing upon more coloured and longer keeping red wines (Dion 1959, Lachiver 1988). In Burgundy, the turnaround is described in a treatise written in 1728 by Claude Arnoux who referred to the emerging distinction between short- and long-keeping wines (Arnoux 1728). Unlike the production of the common "clairet" wines produces in premodern Burgundy, the manufacturing of stronger wines called for harvesting more mature grapes (Lachiver 1988). A similar retardation of mean harvest dates is also known from the region of Montpellier and Beziers in the same period (Blanchemanche 2009).

### 3. Methodology

The new Beaune GHD series was first compared with other grape harvest series as well as with other climate-related proxy time series to test the robustness of the series. Then it was used to reconstruct April-to-July temperature back to 1354. Furthermore, years with extremely early GHD in Beaune were analysed climatologically, with a special emphasis on atmospheric blocking.

### 3.1. Comparison with other time series

The quality of the improved series was first of all tested involving the long Swiss GHD series from 1444 to 2012 (Wetter and Pfister 2013) and the long series of the Czech Lands (from 1499 to 2012) (Mózny et al. 2016). Additionally we have investigated the similarity between the wine harvest dates and tree-ring based temperature reconstructions. Two tree-ring based temperature reconstructions are chosen from the N-TREND data set (Wilson et al. 2016). This is a global data set,





containing the best tree-ring based temperature reconstructions, selected by experts in the field. The spatially closest reconstructions in N-TREND are from the Spanish Pyrenees (Dorado Liñán et al. 2012) and from the Swiss Alps (Büntgen et al. 2006).


The first part of the Beaune series (1354 to 1431) was compared with estimated April-to-July temperatures in Norfolk (South-East England) obtained from the first-time dates of wages paid to grain harvest workers (Pribyl 2017). The second part of the series was compared with the detailed GHD obtained for the period 1420 to 1537 in Metz (France) (Litzenburger 2011). The last part was correlated with the estimated April-to-July temperatures in Central Europe from 1500 to 1759 that are based on Pfister Indices from Germany, the Czech Lands and Switzerland (Dobrovolný et al. 2010). Likewise, the Beaune GHD were compared with the series for Switzerland and the Czech Lands (Fig. 6).


### 3.2. Statistical models

We compared the GHD series not only with indirect climate-observations, but also directly with temperature. Both forward (i.e. reconstruction of GHD from temperatures) and backward (i.e. reconstruction of temperatures from GHD) reconstruction models were used. Correlating GHD month-by-month with Paris mean temperature from 1659 to 2018 (Rousseau 2009, Rousseau 2013, updated to 2018; available on "Euro-Climhist" data-plateform) we found statistically significant correlations between GHD and temperature for all months from March to September. Enologically, using August and September is questionable, in line with the results of stepwise regression analysis attempted by Legrand (1979), Pfister (1984) and Guerreau (1995), all the more as in Beaune harvest sometimes starts already in late August. Therefore we excluded August and September.



We modelled harvest dates in Beaune from Paris monthly temperature, starting in 1659, using two multiple linear regression models relating GHD to temperature in March, April, May, June and July, and fitted with ordinary least squares. Model A is linear while model B used transformed variables. Since the relation between grape growth and temperature is assumed to be non-linear, all temperatures were logit transformed in that model such that curves flatten out for very low and very high temperatures and have the steepest slope at 18 °C (the same transformation was used for all calendar months):


$$T' = \frac{1}{1 + e^{\frac{-(T - 18°C)}{3°C}}} \qquad (1)$$

Further, the harvest date was log-transformed such that a difference in harvest date of one day obtains more weight at the beginning of September than at the end of October:




$$D' = \log(D - 150) \tag{2}$$

Both models were calibrated in the period 1659-1850, while the period 1851-2018 was used for evaluation.

For analysing atmospheric circulation conducive to early harvest dates, climate simulations can be used if they are able to reproduce harvest dates. We therefore applied both models (without recalibration) to the global monthly ensemble climate reconstructions EKF400 back to 1603, which is based on assimilating instrumental data, documentary data and tree ring proxies into an ensemble of 30 global climate model simulations (Franke et al. 2017) as well as to the underlying model simulations CCC400 (Bhend et al. 2012). The closest grid point to Beaune was extracted, and the data were debiased according to the calibration period mean value of each calendar month.

April-to-July mean temperatures were reconstructed from GHD using two models, again calibrated in the period 1659-1850 and evaluated in 1851-2018. Model A is a simple linear regression, yielding the relation:

$$T_{AMJJ} = 36.5[°C] - 0.080[°C / day] \cdot GHD[days] \tag{3}$$

Additionally we defined a model B starting from the simulated GHD in of the CCC400 simulations. We simulated all preindustrial years (1603-1850; before the phylloxera outbreak that destroyed European cultivars and the subsequent reconstruction of the vineyard with American rootstocks) in the 30 member ensemble after debiasing, yielding 7440 years. We assume that any of these years may serve as an analogue provided its GHD is close to the observed one. The reconstruction $x_{rec,j}$ for year $j$ is then:

$$x_{rec,j} = \frac{\sum_i w_{i,j} \cdot x_{m,i}}{\sum_i w_{i,j}} \tag{4}$$

where $i$ refers to the 7440 model years, $x_{m,i}$ is the variable to be reconstructed (here Apr-Jul mean temperature, but the same equation could be used for other variables or other definitions of the warm season) in the model simulation for year $i$ and $w_{i,j}$ is the weight of model year $i$, taken as the density of a normal distribution $N\left(GHD_j, r_{EKF}^2\right)$ at the location of the modelled $GHD_i$. $GHD_j$ is the observed GHD and $r_{EKF}$ is the standard deviation of EKF400 residuals (see above).



### 3.3. Atmospheric circulation blocking

260     For addressing anomalous atmospheric circulation causing early or late harvest dates, we analyse 500 hPa geopotential height (GPH) and atmospheric blocking. The blocks were defined based on 500 hPa GPH according to the algorithm of Tibaldi and Molteni (1990) (see also Tibaldi et al., 1994, Scherrer et al. 2006 for more details). For the period after 1850 we use version 2c of the "Twentieth Century Reanalysis" (20CRv2c, Compo et al. 2011). It provides an ensemble of 56 realisations of an atmospheric reanalyses with 6-hourly time steps. For analysing blocking for early harvest dates prior to 1850, we used CCC400. The performance of blocking algorithms for 20CRv2c and CCC400 has been evaluated in Rohrer et

265     al. (2018). For addressing the latest summer season (2018), we used the ERA5 reanalysis (Hersbach and Dee, 2016).
500 hPa GPH and blocking was reconstructed using equation (4), with $x_m$ denoting Apr-Jul mean fields of 500 GPH and blocking, respectively. This gives atmospheric circulation statistics that are consistent with the corresponding harvest date.

### 4. Results and discussion

### 4.1 Presentation of the Beaune GHD series

The 664-year-long Beaune series is quite homogeneous showing nearly identical averages and standard deviations over the four sub-periods mentioned in section 2 prior to 1988 (Table 3). It is significantly correlated at r = -0.76 with April-to-July mean temperatures in Paris over the 360-year-long period 1659-2018 (Table 2).

The curve is clearly divided in two parts. Grapes were on average picked on 28 September from 1354 to 1987 comprising most of the Little Ice Age and a large part of the 20<sup>th</sup> century. In contrast, GHD's were 13 days earlier (15 September),

during the last 31-year-long period of rapid Global Warming from 1988 to 2018. The main phenological phases in the development of grapes (Vitis vinifera) – bud break, flowering, veraison (change colour and softening of the berries) went in step with the harvest dates (Jüstrich 2018).

Besides the climatic rupture in 1988 several warm (positive) and cold (negative) fluctuations stand out: GHD were 6.5 days earlier between 1383 and 1435 than between 1354 and 1382. The Gorner glacier (Canton Valais, Switzerland) advancing

since the 1340's culminated in 1385 on its first Little Ice Age maximum that corresponds to the position of the glacier in 1859. Then the glacier melted back to a low level which cannot exactly be established (Holzhauser 2010). Likewise the curve mirrored the well-known 1520-1560 and 1720-1739 warm phases as well as the cold c.1600, c.1640 and 1820-1860 phases documented through the waxing and waning of Alpine glaciers (Zumbühl and Nussbaumer 2018). In 1520-1560 grapes were in average harvested four days earlier (24 Sep) than the mean value prior to 1988, and six days earlier (22 Sep)



in the period 1720-1739. While between 1820 and 1860 GHD occurred 4 days later (2 Oct.) than the mean value. In contrast, the rapid warming over the last 30 years accelerated the transition from a melting-back remelting process to an actual decay.

## 4.2 Correlation with other proxy time series

Correlations between documentary-based proxy series and the Beaune GHD series turned out to be significant (Fig. 6). We focused on the period prior to 1850 and significant Pearson correlation was found between Wheat Harvest Dates in Norfolk
and Grape Harvest Dates in Beaune despite the considerable distance between the two locations and the different nature of the proxy. The GHD available for Metz from 1420 to 1537 are well correlated to Beaune GHD as well (r = 0.60). On the other hand the coefficient is surprisingly high between the GHD from the Czech Lands and the Beaune series despite the distance of 900 km between Beaune and the region northwest of Prague.

Tree-ring based temperature reconstructions were used to investigate the similarity with the grape harvest dates. Both have
annual resolution and are influenced by the summer growing season. Similar to the long instrumental measurements from Paris, selected tree-rings reconstructions recorded temperature variations at a distance of a few hundred kilometers from Beaune. Nevertheless, seasonal average temperatures should be highly correlated over regions of several hundred kilometers. Pearson correlation coefficients are expected to be negative because the warmer a growing season is, the earlier the harvest date and the thicker the tree ring or the denser the latewood. Correlation coefficient for both sites are high and clearly
significant (p < 0.05), obviously not as high as with instrumental temperature because tree-ring proxies include additional noise from non-climatic influences. Correlations coefficient are robust and remain constant throughout all the tested sub-periods (Table 4).

## 4.3. Statistical model for harvest dates

GHD can be well reconstructed from temperature, using either model A or B (see Table 5, Fig. 7; scatterplots are shown in
Fig. S1). Interestingly, both models produce a better correlation (>0.8) in the validation period than in the calibration period, probably due to the strong and well reproduced trend of GHD in the validation period. Given the fact that the observed temperature refers to a location more than 300 km away and is based on early instruments with presumably substantial errors, a correlation coefficient of 0.8 over the entire period is indeed surprising.

Correlations are still high when applying model B to EKF400 (Table 5 top shows the range over the entire ensemble; see
also the scatter plots in Fig. S1). This suggests that GHD can be modelled from climate simulation output, which is important for the analogue reconstruction approach.

The reconstruction of April-to-July mean temperature from GHD using models A and B (analogue resampling of CCC400) yields similar statistics as the forward approach (Table 5, Fig. 7, bottom). In the observations, April-to-July temperature



reached the highest value ever in 2018. No year in the past was warmer in the observations (although there were three years with earlier harvest).

**4.4. Role of atmospheric blocking**

What atmospheric conditions are conducive to early GHDs? Using the analogue approach we can analyse the April-to-July averaged 500 GPH and blocking statistics over the North Atlantic-European region for past years. Figure 8 shows GPH (anomalies in contours) and blocking (anomalies in colour, climatology in contours) fields that are consistent with the GHD

of 1556; the second earliest on record after 2003. For comparison, we also show a composite of summer blocking for the 10 earliest GHDs in the period 1851 to 1980 from 20CRv2c reanalysis (we excluded the last decades due the strong anthropogenic warming effect) relative to the average over that period. Finally we also show blocking anomalies for summer 2018 relative to the average 2000-2018 from ERA5 reanalysis.

Early GHDs are related to high-pressure anomalies centred over Western or North Central Europe. High pressure situations

are accompanied by increased radiation and high temperatures. With respect to blocking, anomalies are weak over the study area (central-western Europe). Rather, blocking during early GHD years occurred more frequently over Denmark (less frequently over Northern Scandinavia). In such situations the study area lies to the southwest of the block and receives dry and warm continental air masses. A similar blocking pattern is also found for the 10 earliest GHD years in the 20CR reanalysis. The year 2018 follows a similar pattern. Early harvest dates are thus related to blocking over Denmark. Late

harvest date (not shown) do not imprint significantly onto blocking.

**4.5 Grape ripening in extremely hot and dry years**

The 33 extremely warm events comprising the 5% percentile bracket of GHD are unevenly distributed over time (Fig. 9). 21 of them occurred between 1393 and 1719, i.e. one out of 15 years included a hot spring-summer period. In contrast, this is the case for just five years between 1720 and 2002, i.e. one out of 56. Under those circumstances, the memory of

outstandingly warm years faded. No wonder that the hot summer 2003 came as a surprise. Since then 8 out of 16 spring-summer periods were outstanding according the statistic of the last 664 years, no less than 5 among them within the last 8 years. This implies that the extremes in the past have now become normal. The acceleration of extreme temperatures in the last decade went along with an increased melting back or decay of Alpine glaciers which lost about 20% of their remaining volume (Swiss Glaciers 2017).

Considering the extremes prior to the onset of rapid Global Warming, it is striking that the GHD in the outstandingly warm and dry spring-summer period in 1540 only ranks 19[th] in the statistic of the earliest years. In contrast, the GHD in 1556 ranks second coequal with 2018 just after 2003, though conditions in 1556 are not as frequently highlighted in chronicles as in



1540. An interpretation of this paradox is attempted using detailed vine phenological evidence available from vineyards around Biel-Bienne, Zürich and Schaffhausen (Switzerland) in 1540 and 1556 in comparison with the Beaune evidence (Table 6). Source references are provided in the Euro-Climhist data-platform. Elevation matters for the comparison: the three Swiss vine growing areas are located at altitudes of about 430 metres asl, i.e. about 130 to 200 metres higher than those of Beaune region vineyards. Vine cultivars (CV), i.e. varieties, need to be considered besides altitudes. In Biel-Bienne CV Chasselas used to be grown while in Zurich and Schaffhausen CV Räuschling was cultivated which survives in a few vineyards until today. The maturity of the Chasselas and Räuschling cultivars is 10 to 14 days earlier than that of Pinot noir grown in the Beaune area.

Table 6 lists the main phenological grapevine stages and harvest dates for years 1540, 1556, 2003 and 2018. In Switzerland, the three stages 'end of flowering', 'beginning of veraison' (i.e. the stage during which grape coloration and softening occur, usually considered as the beginning of the grape ripening period) and 'first ripe grapes' occurred nearly on the same days in 1540 and 1556, whereas large differences appear between the stage of veraison and the harvest date in both years. On average the difference between these two stages is rather constant fluctuating between 35 and 40 days (Daux et al. 2012). Gladstones (2011) refers to the widely observed fact that "the date of flowering, can usually predict quite closely the dates of veraison (colour change and softening of berries) and maturity to follow [...]. The later phenological intervals show little response to temperature and tend to be constant from year to year". His assessment is confirmed by Chuine et al. (2004) and Jüstrich (2018). Since the early 21st century, higher temperatures combined with increase control of grape sanitary status (grey mould disease mostly) makes ripening duration (i.e. the lag during veraison and harvest) more winemaker dependent. The winemaker choices depend on both cultivar and style of wine. For instance, the number of days between veraison and harvest for CV Cabernet-Sauvignon has nearly been doubled in a famous Chateau in the appellation Margaux near Bordeaux (Van Leeuwen and Destrac-Irvine, 2017). In 1540, however, grapes in the Zürich area were harvested 56 days after veraison, i.e. with a substantial delay. The hot and dry period in 1540 lasted from April to the end of the year. The heat-wave peaked at the end of a 46 day long rainless period between 23th June i.e. three weeks before veraison and 7th August during which many forests and settlements in a large area from the Ardennes to Poland got up in flames (Pfister 2018). Maximum temperatures from late July may have exceeded 40°C (Orth et al. 2016). Oenological research established that under conditions of extreme heat and drought the development of grapes is slowed down or stopped (Keller 2016). When occurring before veraison, extreme hydric stress alters grape quality and the onset of ripening, possibility because it induces leaf defoliation and therefore carbon assimilation limitation (Basile et al., 2012; Girona et al., 2009; Ollat and Gaudillere, 1998)). In fact, this phenomenon was observed during the temperature peak in the hot summer 1947 in Schaffhausen (Amtsblatt 1947). Besides a delayed grape maturity it seems that in 1540 the vintage ban was not observed any more. Vine-growers in Schaffhausen were waiting in vain for rain to begin the harvest, as chronicler Oswald Huber relates, which suggests that



precipitation rather than temperature finally mattered. However, vine-growers finally tackled the work nevertheless, because the plants withered (Wetter and Pfister 2013).

The 1556 harvest of Chasselas grapes in Biel-Bienne was estimated to have occurred between 25th and 30th August i.e. at about the same time as that of Pinot noir in Beaune considering the chronicler's remark that the abundant harvest already ended on 10th September (Gregorian Style). This leads to conclude that the harvest in 1556 began between 52 and 57 days after veraison that is to say with about the same delay as in 1540. The preceding winter 1556 had been very wet considering the daily weather observations by Wolfgang Haller in Zürich. Disregarding June, which was completely rainless, the spring-summer period in Zürich included 7 precipitation days in April, 3 in May and 5 in July (Haller in Pfister, Rohr https://echdb.unibe.ch/, 15th Oct 2018). A hot and almost rainless period began on 29th July and lasted to 11th September, i.e. more than a month later than in 1540.

In central France, the 1556 heat and drought began in mid-April, i.e. about 45 days later than in 1540, likewise following a wet winter. The heat-wave from mid-April to mid-June led the vegetation quasi explode. In the Loir-et-Cher region, a variety of red Pinot noir cultivar named Auvernat was already blossoming around the 25 of April (Nouel, 1878, 235). Not a drop of rain felt during this period. On 14th June, it was sheeding down for three to four hours which visibly refreshed the vegetation. Before 10th July the first grapes were ripe. In July the ground became so hot that it burnt people's feet walking barefoot. Like in 1540 the heat and drought peaked in early August (Bourquelot, 1857, 30-31; Hiver 1867, 81 and 90; Nouel 1878, 235-237). Following the deliberation protocols of the chapter of Notre-Dame of Beaune, seven processions for obtaining rain were held in the city from 15th June to 15th August (Arch. Dep. Côte d'Or, G 2499).

In 2003 the water deficit was not sufficiently strong to "block" grape ripening through extremely limited photosynthesis. Obviously, ripening made is course quickly. Additionally, wine growers harvested pretty soon to maintain sufficient acidity in wines mostly (and in some cases avoid excessively high alcohol content in wines).

In sum, it is concluded that the decline of 13 days in the average date of GHD since 1988 went along with large increase in the number of warm spring to summer seasons. Moreover it seems that temperatures from outstanding GHD are rather underestimated due to sensibility of grapes to heat and drought stress.

**5. Conclusions**

Time series of documentary proxy data such as GHD need to be critically evaluated by historians prior to statistical analysis. 19th century's publications need to be cautiously examined before being used and in any case first-hand documentary material should be preferred, which takes a lot of meticulous detail work in the archives.



The 664-year-long Beaune GHD series assembled from the archives of the city is significantly correlated with the long Paris temperature series from 1659 to 2018 and with documentary and tree-ring proxy data from 1354 to 1658. Statistical models describe Beaune GHD very well, with Pearson correlations around 0.8. The climate rupture of 1988 divides the series in two

different parts. Over the period of the Little Ice Age and the "warm twentieth century" up to 1987 grapes in the Beaune area were picked on 28 September on average. After the climate rupture in 1988 the harvest date declined by 13 days to an average of 15th of September during the last 31-year-long period of rapid Global Warming from 1988 to 2018. It is noteworthy that the 33 values below the 5% percentile are unevenly distributed over time. While 21 of them occurred between 1354 and 1719, only 4 of them were registered between 1720 and 1987. In contrast, 8 outstanding extremes

occurred within the last 30 years, 3 between 2000 and 2010 and 5 between 2011 and 2018, which probably witnessed the warmest warm season temperatures since 1354.

Early harvest dates coincide with high pressure influence and increased blocking over Denmark. Conversely, the most outstanding heat and drought years were not necessarily the earliest in the ranking of harvest dates. It is concluded that grape development slowed down or even stopped during very long rainless periods and extreme maximum temperatures such as in

1540 and 1473 (Camenisch et al. in preparation). In sum, the long homogenized Beaune series visually demonstrates that warm extremes in the past were outliers, whereby they have become the norm in the present time.

**Author contribution:**

Thomas Labbé provided the Beaune GHD dataset by archival researches, and wrote section 2 and 4.1.

Christian Pfister wrote section 2 and 4.1 together with Thomas Labbé and wrote sections 4.2 and 4.5.

Stefan Brönnimann and Jörg Franke performed the statistical reconstructions and wrote sections 3.2, 3.3, 4.3, 4.4.

Daniel Rousseau provided the updated Paris mean temperature series (1659-2018) and wrote section 2.6.

Benjamin Bois provided information on vine phenology and helped writing section 4.5.

**Competing interests:**

The authors declare that they have no conflict of interest.

**Acknowledgments**:

Stefan Brönnimann and Jörg Franke were supported by the Swiss National Science Foundation (project RE-USE) and by the

European Commission (ERC Grant PALAEO-RA). Simulations were performed at the Swiss National Supercomputing Centre CSCS. The Twentieth Century Reanalysis Project datasets are supported by the U.S. Department of Energy (DOE) Office of Science Innovative and Novel Computational Impact on Theory and Experiment (INCITE) program, and Office of



Biological and Environmental Research (BER), and by the National Oceanic and Atmospheric Administration Climate Program Office

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





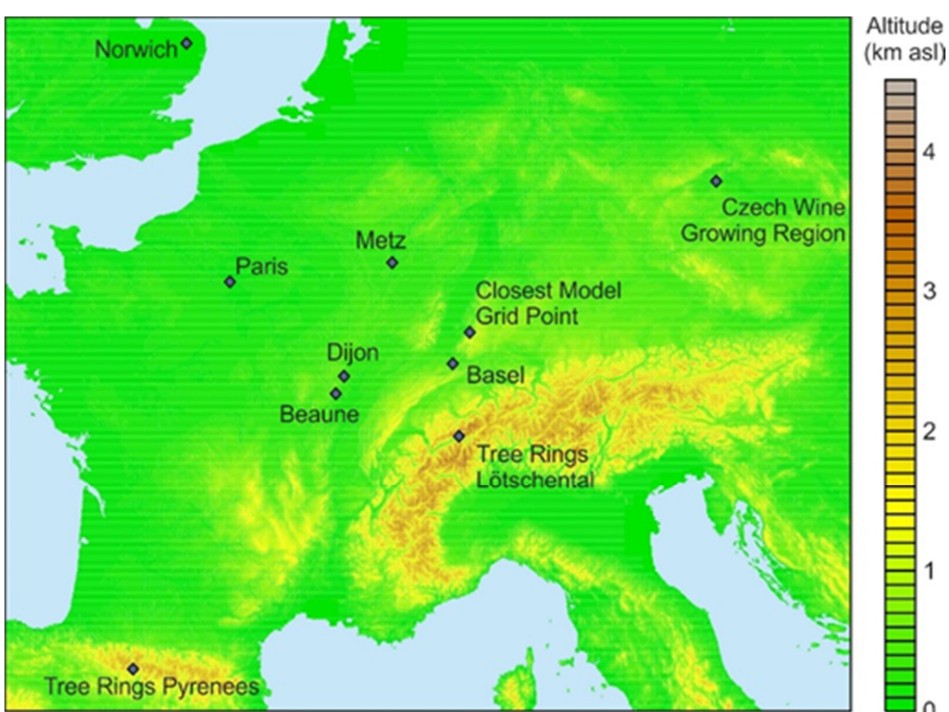

*Figure 1: Geographical area oft he study.*




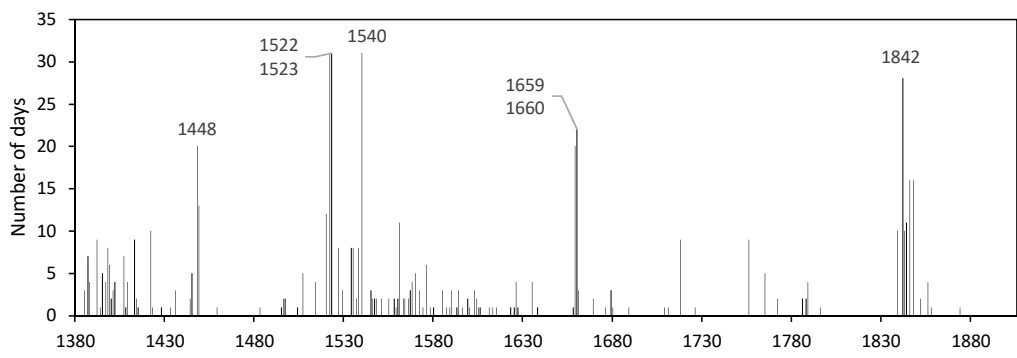


*Figure 2: Mismatches between "Angot 1885" and "Labbé-Gaveau 2011" Dijon GHD series (1385-1905).*




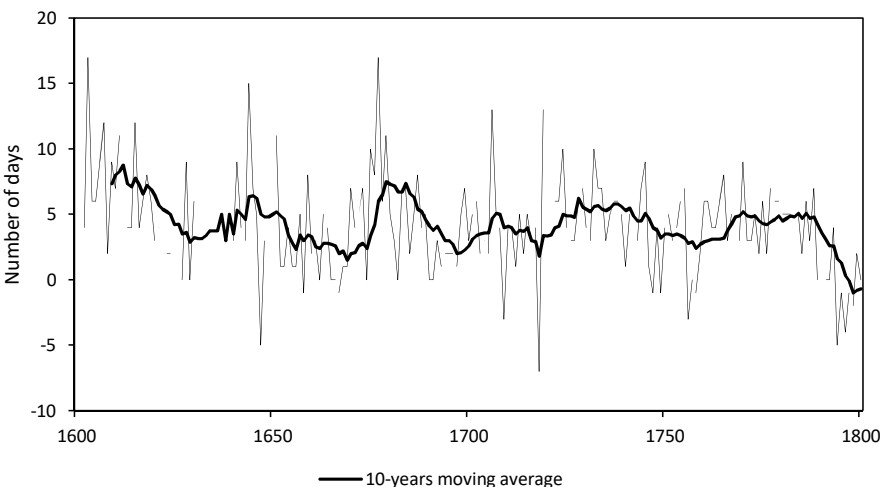

*Figure 3: Time interval between the GHD of Dijon and the GHD of Beaune (1600-1800).*



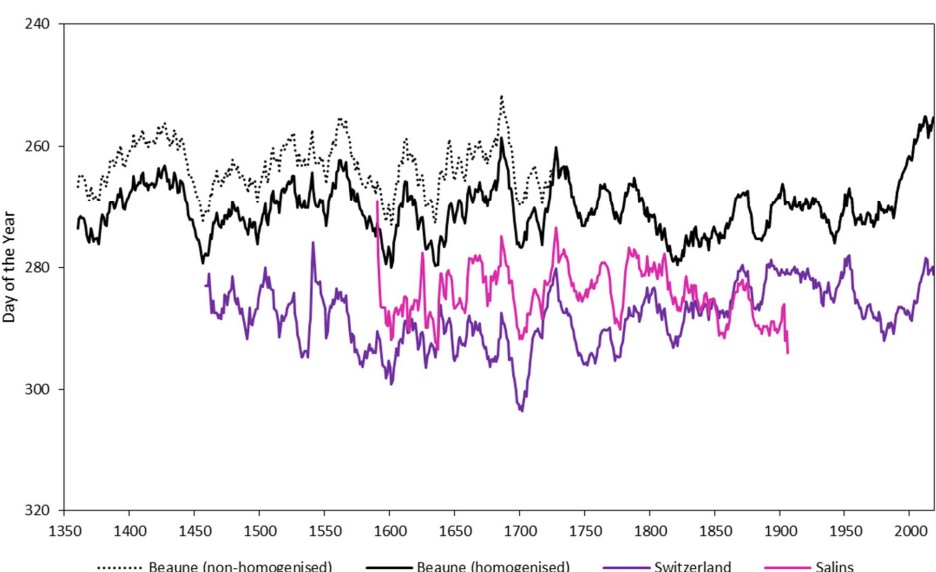

*Figure 4: 11-years low- pass filtered GHD series of Beaune (homogenised and non-homogenised), Switzerland (Wetter et al. 2013) and Salins (Angot 1885)*







*Figure 5: Relation between April-to-July mean temperature (°C) in Paris and Beaune GHD. Green dots indicate the calibration period, blue dots evaluation period. The linear regression line is shown.*





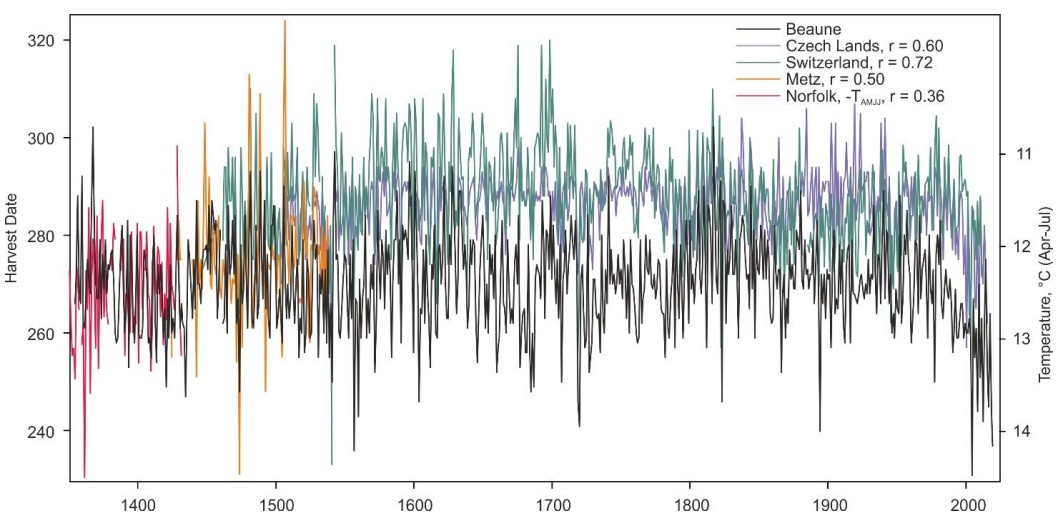


*Figure 6: Comparison of times series of GHD from Beaune (this work), Czech Lands (Mózny et al. 2012), Switzerland (Wetter et al. 2013), and Metz (Litzenburger 2011), as well as temperature reconstructions for Norfolk (Prybil et al. 2012; right scale, inverted). The number indicate the correlation between the series prior to 1850.*




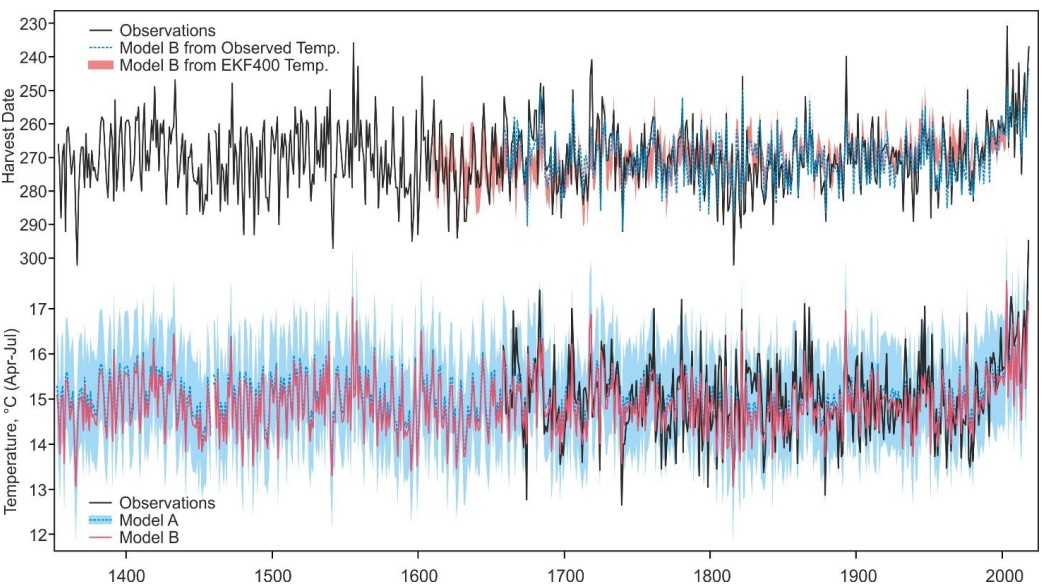

*Figure 7: (top) Harvest dates in observations and modelled from observed and reconstructed (EKF400) temperatures using model B.*
*(bottom) April-to-July mean temperature from observations and reconstructed from models A and B. Blue shading denotes the 95%*
*prediction interval.*






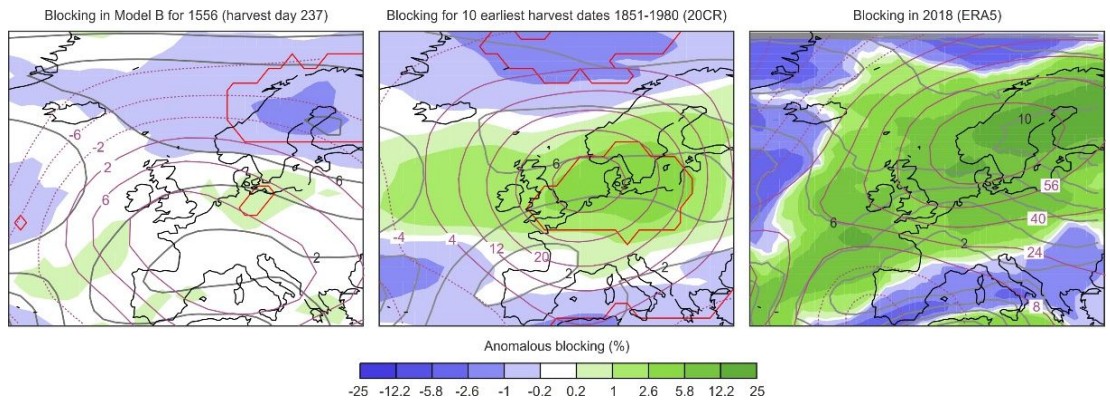

Figure 8: Anomalies in April-to-July 500 hPa GPH (purple contours in gpm) and blocking frequency (in % of time steps blocked, note the non-linear scale) in (left) model B for 1556 (relative to all years prior to 1850) as well as (middle) averaged for the 10 earliest harvest dates in 1851 to 1980 in 20CRv2c (relative to 1851-1980). The right figure shows anomalous 500 hPa GPH and blocking in April to July 2018 in ERA5 (relative to 2000-2018). Grey solid contours give the corresponding blocking climatologies (in % of time steps blocked). Red lines indicate where the 5% to 95% confidence interval for anomalous blocking is exceeded based on 1000 repetitions of a Monte Carlo sampling of the weights (for model B) or years in 20CRv2c, respectively.





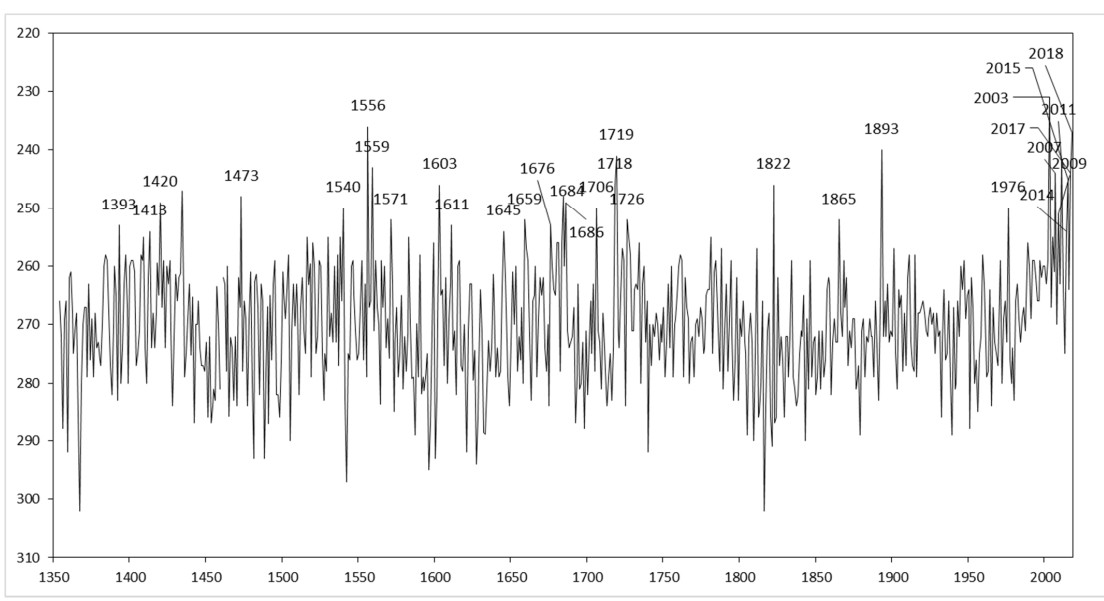

*Figure 9: Beaune GHD time series (1354-2018), with indication of the 5% earliest dates.*






Table 1: Comparison of mean GHD in Beaune, Switzerland and Salins times series.

|  | 1599-1717 | 1718-1875 | 1354-1717 | 1718-2018 |
|---|---|---|---|---|
| Mean Beaune GHD | 21 Sep | 28 Sep | 20 Sep | 27 Sep |
| Mean Swiss GHD | 20 Oct | 15 Oct |  |  |
| Mean Salins GHD | 11 Oct | 09 Oct |  |  |


Table 2: Pearson correlation (r) between Beaune GHD series and March to September mean Paris temperatures (1659-2018).

| March (M) | Apr (A) | May (M) | Jun (J) | Jul (J) | Aug (A) | Sept (S) | AMJJAS | AMJJA | AMJJ |
|---|---|---|---|---|---|---|---|---|---|
| -0.28 | -0.42 | -0.51 | -0.53 | -0.43 | -0.30 | -0.23 | -0.73 | -0.74 | -0.76 |







*Table 3: Mean GHD and Standard deviation for various subperiods of the Beaune GHD homogenised time series.*

|  | **1354-1506** | **1507-1699** | **1700-1965** | **1966-1987** | **1988-2018** |
|---|---|---|---|---|---|
| Mean GHD | 28 Sep | 27 Sep | 28 Sep | 28 Sep | 15 Sep |
| Std. Dev. | 9.88 | 10.60 | 8.91 | 7.20 | 9.93 |


*Table 4: Pearson correlation coefficients (r) for various subperiods between wine harvest dates and tree-ring based temperature reconstructions. All correlations are statistically significant (p<0.05).*

|  | **1354-2010** | **1354-1659** | **1659-2010** | **1711-1770** | **1771-1850** |
|---|---|---|---|---|---|
| **Pyrenees** | -0.41 | -0.36 | -0.44 | -0.48 | -0.48 |
| **Lötschental** | -0.32 | -0.35 | -0.34 | -0.41 | -0.56 |




*Table 5: Reconstruction statistics. A and B refer to the statistical models, subscript Obs. and EKF400 denote the application of the calibrated model to Observations and EKF400, respectively. "Cal" and "Val" indicate the model statistics in the calibration and validation periods, respectively, sd. indicates its standard deviation (in days and °C, respectively) for the common period (1659-2004). Numbers for EKF400 refer to the ensemble mean of 30 reconstructions.*

|  | **Reconstructing GHD from Temperature** | | | | **Reconstructing $T_{Apr-Jul}$ from GHD** | | |
|---|---|---|---|---|---|---|---|
|  | *Obs.* | $A_{Obs}$ | $B_{Obs}$ | $B_{EKF400}$ | *Obs.* | $A$ | $B_{Analog}$ |
| $r_{cal}$ |  | 0.704 | 0.727 | 0.653 |  | 0.704 | 0.707 |
| $r_{val}$ |  | 0.824 | 0.831 | 0.725 |  | 0.824 | 0.820 |
| $bias_{cal}$ |  | 0.00 | 0.00 | 0.62 |  | 0.00 | -0.18 |
| $bias_{val}$ |  | 1.27 | 1.06 | -0.97 |  | 0.03 | -0.15 |
| sd | 9.37 | 6.91 | 7.10 | 6.19 | 0.85 | 0.57 | 0.60 |





715 *Table 6: Phenological stages (flowering, veraison, ripening of the grapes and harvest dates) of the vine during extremely early years. S (Schaffausen, white Räuschling cultivar); Z (Zürich, white Räuschling); Bi (Biel-Bienne, white Chasselas), M (Malans, red Pinot noir), B (Beaune, red Pinot noir)*

| Year | flower (end) | DOY | veraison | DOY | Ripe (grapes) | DOY | GHD | DOY | veraison to GHD (days) |
|------|--------------|-----|----------|-----|---------------|-----|-----|-----|------------------------|
| 1540 | 10 Jun (S) | 162 | 5 Jul (S) | 187 | 16 Jul (Z) | 198 | 11 Sep (Z) | 254 | 67 (S-Z) |
| 1540 | | | | | | | 3 Sep (B) | 247 | |
| 1556 | 10 Jun (Z) | 162 | 4 Jul (S) | 186 | 19 Jul (Z) | 201 | 25-30 Aug (Bi) | 238-243 | 52 -57 (Bi-S) |
| 1556 | | | | | | | 26 Aug (B) | 239 | |
| 2003 | 30 May (B) | 150 | 28 Jul 28 (B) | 209 | | | 20 Aug (B) | 232 | 23(B) |
| 2018 | 31 May (B) | 151 | 31 Jul 31 | 212 | | | 30 Aug (B) | 242 | 30 (B) |
| 2018 | 31 May (M) | 151 | 1 Aug (M) | 213 | | | 17 Sep (M) | 261 | 48 (M) |