# Peer review of "The longest homogeneous series of grape harvest dates, Beaune 1354-2018, and its significance for the understanding of past and present climate"

_Climate of the Past, 2018_

## Short Comment (SC1) · 11 Jan 2019

Thanks for submitting these interesting results. A very minor and short comment. Could you later provide a much better map of the study area than the present figure 1. Very sincerely one cannot get any idea of the correct relief as the color code used gives the idea of a rather flat area which is not the case.

All the very best

denis

---

## Author Comment (AC1) · 21 Jan 2019

Dear Mr. Rousseau, thank you very much for your interest in this work and for your comment. We will actually provide a better map of the study area in the final version. Best wishes, Thpmas Labbé.

---

## Referee Comment (RC1) · Tim Heaton (Referee) · 13 Feb 2019

**1   Summary**

I would like to thanks the authors for letting me review their work. I found it a very interesting read and it is a very impressive dataset. The authors have carefully collected a complete annual history of the grape harvest dates in the Beaune region by collating various archival records all the way back to 1354 AD.

[Figure]

These grape harvest dates are thought to provide strong correlation with temperatures in the growing season (April-July) since the harvest date is linked to the ripeness of the grapes. The authors evidence this correlation through comparison with the Paris temperature series from 1659-2018 where good agreement is seen. This suggests that the Beaune record may be used to provide climate information extendinf further back in time to the 1300s.

The work is well written although it would benefit from some minor English grammar and spelling alterations.

**2   Comments on work**

I have four main comments/suggestions for the authors described below

**2.1   Improved introduction and clarification of notation**

As someone without any previous knowledge of grape harvesting in France it took me until a few pages in (and the first plot) to entirely understand what a GHD was. Initially I wondered if each individual vineyard might have a different harvest date and so you were recording many dates in a single year. For a non-expert, the paper perhaps presumes a little too much prior knowledge in this area in particular regarding the terminology.

Even after reading I was still a little confused as to some of the practical aspects of grape harvesting from the perspective of the physical harvester. How in general is a GHD decided upon? What are the factors that come into play? Presumably ripeness of grape but are there other factors such as weather forecast; national holiday; the weekend/weekday? Who decides on grape ripeness and how do they decide?

[Figure]

Is a GHD the data on which the town administration permit harvesting to begin, but an individual vineyard can harvest after this if they like (but not beforehand)? Or do all vineyards in the region have to harvest on the same date? What is a ban (and specifically a vintage ban)? is it that all vineyards are banned from harvesting before a certain date; if so how is this different from a GHD? All these terms were used in the paper without my full understanding.

It would be nice if the authors could include a short (imagine 1-2 paragraphs would be sufficient) description at the start to give a bit of a practical background to grape growing and harvesting in France; a description of a typical cyclical year in a vineyard; and a description of all the later terminology.

**2.2 Consistency of determining the GHD**

I am particularly impressed by the work that has obviously gone into collating the GHDs from different historical records; and in particular the clarity and honesty in describing the relative strengths/ weaknesses of the four approaches. Of the four approaches the one requiring the strongest assumptions is that based upon Church records (1507-1699). Here the authors have assumed that the GHD takes place 8 days after the church meetings to organise food provisions.

The accuracy of this delay is critical in determining the level/amount of harmonisation required for the series. If, it in fact took 15 days (i.e. an extra week) from the church meeting until the GHD then this would mean no harmonisation of the time series would be required (or at least not at 1700 but rather more likely much earlier).

The paper suggests there are some years for which ones has both the dates of church meetings; and either wage payment or city council records albeit potentially fragmentary. It would be good to see what the delay between church meetings and GHD was for these years when the GHD was known exactly to see if this delay is consistently 8

years; or if it is smaller/larger.

**2.3 Harmonisation of time series**

The issue which, in my mind, is likely to cause most discussion/disagreement amongst readers is the harmonisation of the time series whereby the raw GHDs from before 1718 are moved 7 days later in order to make the series as a whole agree with other GHD records. The authors do provide what appears to be a reasonable justification for this — a change in wine production and commercialisation. Without a specific administrative edict describing a decision to call the GHD in the region later as a consequence this does however slightly weaken the series as an entirely independent way to tell us about climate in the region — do we know for sure that there are not other similar changes at other times in history? However, the fact this is a uniform and constant shift does alleviate this concern to a large extent in my eyes.

This made me wonder somewhat as to whether the primary interest/power in the series was more to do with climate or rather to identify changes in vinoculture practices. Equally, the fact this change occurs at 1718 is presumably heavily reliant upon the assumption of an 8 day delay between church records and the GHD. If the delay were longer then the shift would be applied at a different cut-off (e.g. much earlier in time).

**2.4 Statistical modelling**

As a statistician, I found the details of the modelling section 3.2 a little hard to follow in terms of exactly what models were fitted. It would seem from the description that two model approaches were used — a frequentist approach in model A; and potentially a Bayesian model B using CCC400. If this is correct then I suggest below a way the authors might wish to describe the models more clearly (and in more formal statistical language) so they could be easily reproduced/understood by others:

**2.4.1 Frequentist Approach — Model A**

Here I was confused with where the transformation of the variables occurred. As some-time the authors introduced a $T'$ and $D'$ but then their models as described include neither of these terms. Can they just write down the model in terms of the notation they define earlier? Is the model for example

$$T' = 36.5 - 0.080D'$$

where

$$T' = \frac{1}{1 + \exp\left(\frac{-(T-18)}{3}\right)}$$
$$\text{and} \quad D' = log(D - 150)?$$

Also, is $log$ natural log or base 10?

**2.4.2 Bayesian Approach — Model B**

If you wanted to frame model B as being a Bayesian method then this could be done as follows. The CC400 simulations provide a prior on $x_j$ (the April-July mean tempera-ture). This prior is in terms of equally-likely sample values $(x_{m,i})_{i=1}^{7} 440$, each of which is accompanied by a modelled $y_{m,i}$ (the corresponding GHD for $x_{m,i}$). We wish to find a posterior estimate for the April-July mean temperature in light of the true GHD $z_j$ by reweighting the prior according to Bayes theorem:

$$P(X_j = x_j | z_j) \propto f(z_j | x_j) P(X_j = x_j)$$

where $f(z_j | x_j)$ is the conditional density of $z_j$ given $x_j$ i.e. $w_{i,j} = f(z_j | x_j) \sim N(z_j; x_j, r_{EKF}^2)$, the density of a normal with mean $x_j$ and variance $r_{EKF}^2$. Our reconstructed temperature is then the posterior mean of $X_j|Z_j = z_j$:

$$E[X_j|Z_j = z_j] = \frac{\sum_i w_{i,j} x_{m,i}}{\sum_i w_{i,j}}.$$

2.5  Possible suggestions the authors might wish to consider for future work?

- While not expected in this paper, one might wish to formalise the identification in changes to the time series using changepoint methods. This could identify both changes in climate but also potentially vinoculture practices (in light of the observed change in grapes around 1718). For example is there a point at which irrigation became more common/advanced that affected the GHD? There are some off-the-shelf (and hopefully fairly easy to apply) statistical packages which might provide insight that visually is harder to see.

---

## Referee Comment (RC2) · Anonymous Referee #2 · 7 May 2019

The manuscript 'The longest homogeneous series of grape harvest dates, Beaune 1354-2018, and its significance for the understanding of past and present climate' by Thomas Labbé and co-authors is well-written, logically structured and highly relevant, as it provides a critically thorough and greatly advanced perspective on the paleoclimatic potential and limitations of grape harvest dates (GHD).

The new GHD record from Beaune is a true improvement of previous GHD-based warm-season reconstructions from both, France and Switzerland, because of the data used and methods applied.

It is particularly interesting that the new reconstruction does not corroborate the putative heatwave in 1540 CE, which is broadly in line with European-wide tree-ring evidence (Büntgen et al. 2015); an aspect that should be discussed in a more balanced and nuanced way than currently done.

Before seeing the paper published (to which I am very much looking forward), I kindly ask the authors to consider the spatially explicit European summer temperature reconstruction by Luterbacher et al. (2016) for comparison (or provide reasonable argumnents for why not using this state-of-the-art high-res product). Moreover, the article would benefit from an assessment of the persistence (i.e. autocorrelation structure) in Beaune's new GHD-based April–July temperature reconstruction compared to the possibly inflated long-term persistence in tree ring width records – a research frontier at least some of the authors are very familiar with (e.g. Franke et al. 2013).

In summary, this is study will make a significant contribution to the field of high-resolution paleoclimatology, and I applaud the authors for what they have accomplished in terms of data, methods and interpretation.

---

## Author Comment (AC2) · 27 May 2019

Dear Tim Heaton, we thank you for having taking time to carefully read our paper and for the valuable propositions of improvements you suggest in your comment. In the next lines we try to answer to all points discussed in your review.

**1. Improved introduction and clarification of notation**

The analysis of GHD times series is since the end of the 19th century a long-lasting and very well-known method of investigation in the field of historical climatology. Accordingly, all aspects of the organisation of the wine harvests have been fully discussed in the previous literature (Lavalle 1855, Dufour 1870, Le Roy Ladurie 1971, Labbé-Gaveau 2011, Daux et al. 2012). In our article, section 2.4 provides a fully detailed description of the organisation of the harvest in the region of Beaune since the end of the Middle Ages. Considering this, and the fact that we have to limit the length of our text to the standard maximum of the review, it would be superfluous to add one or two paragraphs describing the harvest process in details. Thought, we agree that there can be some difficulties to interpret the meaning of certain terms, like the distinction between" GHD" and "ban dates" for example, and that some clarifications are needed. We have then improved the introduction in that sense.

**2. Consistency of determining the GHD**

This is a critical point concerning the setting of the series. You rightly point out that the period 1507-1699, for which Beaune GHD are extracted from deliberation protocols of the church of Notre-Dame of Beaune, constitutes the less reliable part of the series. In this period we actually had to estimate for each year a "probable" opening date of the harvest, for documentary records of the church provide only the date of the meeting in which the estate managers of the institution had to anticipate the upcoming harvest. Before 1582, as the given date is the date of the last meeting before the vacancy of the canons chapter for the harvests, we add three days to this dates, i.e. the standard interval between two meetings. After 1583, the protocols inform on the organisation of food supplies for harvesters, so that we added 8 days to this date. The accuracy of these estimations, though we must admit some uncertainties, is confirmed with the comparison of fragmentary preserved official ban dates set by the city of Beaune for some few years within this period, as summarized in Table 1 :

*Table 1 : Comparison between dates of the last meeting of the Notre-Dame of Beaune church chapter before the grape harvest and the official ban harvest set by the city council (1554-1620)*

|  | 1554 | 1555 | 1557 | 1558 | 1569 | 1574 | 1583 | 1613 | 1617 | 1619 | 1620 |
|---|---|---|---|---|---|---|---|---|---|---|---|
| Ban dates set by the city council of Beaune | 3 Sep | 19 Sep | 7 Sep | 6 Sep | 9 Sep | 13 Sep | 5 Sep | 21 Sep | 27 Sep | 20 Sep | 25 Sep |
| Dates given by the protocols of the church of Notre-Dame | 23 Aug | 13 Sep | 3 Sep | 2 Sep | 2 Sep | 15 Sep | 1 Sep | 18 Sep | 20 Sep | 6 Sep | 16 Sep |
| Estimated dates | 26 Aug | 16 Sep | 6 Sep | 5 Sep | 5 Sep | 18 Sep | 8 Sep | 26 Sep | 28 Sep | 14 Sep | 24 Sep |

This table will be provided as supplementary material in the final version of the article. Furthermore, it must be noticed that when an official ban dates set by the city council of Beaune was available in the documentation, we always integrate this date in the series.

**3. Harmonisation of time series**

The harmonisation of the time series before and after 1718 is also a critical problem. The choice to add 7 days to all dates of the Beaune GHD time series before 1718 is based upon the observation that the mean Beaune GHD is 20[th] of September in the period 1354-1717 and 27[th] of September in the period 1718-2018 (see Tab 1, discussion paper). We suggest that this uniform delay reflects an anthropogenic changes affecting the setting of the harvest date. Otherwise it would have mean AMJJA temperature reconstructions with maxima and minima c. 1°C warmer before 1718 than afterwards on the decadal scale, even though temperatures measurements available in France since 1658 do not provide any evidences of such a warming during the period 1658-1718. Moreover, extra-regional GHD times series (Switzerland, Jura, Czech Lands) during comparable time span (1599-1875) shows different patterns of evolution. On the one hand all series are more stable, and we observe on the other hand a general trend for later, and not earlier, GHD in the pre-XVIII[th] century period (Tab. 2).

*Table 2 : Mean GHD in Beaune (this article), Aubonne (Angot 1885), Salins (Angot 1885), Switzerland (Wetter et al. 2013) and Czech Lands (Mózny et al. 2016).*

|  | Period 1: 1599-1717 | Period 2: 1718-1875 | Difference of days between period 2 and period 1 |
|---|---|---|---|
| *Beaune mean GHD* | 21 Sep | 28 Sep | +7 |
| *Aubonne (Jura) mean GHD* | 21 Oct | 18 Oct | -3 |
| *Salins (Jura) mean GHD* | 11 Oct | 9 Oct | -2 |
| *Swizz mean GHD* | 20 Oct | 15 Oct | -5 |
| *Czech Lands mean GHD* | 14 Oct | 14 Oct | 0 |

We assume first that the more unstable time series is the one which have to be homogenised. Secondly it appears that we should logically add some days to the Beaune GHD time series before 1718. We acknowledge however the hypothesis that this rupture is approximatively synchronous with a change in our documentary sources, i.e. the changeover from protocols of the Church to the deliberation of the city council of Beaune. It is actually a matter of discussion to what extent this rupture reflects a documentary bias or not.

Two arguments can nonetheless alleviate this concern. First, the fact that mean Beaune GHD was earlier in pre-18[th] century period is observable in the 14[th] and 15[th] centuries as well (Tab. 3). For this period, the data have been collected up to 1507 from a third kind of archival material, i.e. the accounting documentation of Notre-Dame estate, providing direct information about the exploitation of the vineyard and then not questionable regarding their reliability. It confirms then the reliability of the protocols used for collecting 16[th] and 17[th] centuries GHD.

*Table 3 : Evolution of non-homogenised mean Beaune GHD over centuries*

|  | 14[th] c. | 15 th c. (until 1507) | 16[th] c. (after 1507) | 17[th] c. | 18[th] c. | 19[th] c. | 20[th] c. |
|---|---|---|---|---|---|---|---|
| Mean Beaune GHD (non homogenised) | 22 Sep | 20 Sep | 20 Sep | 20 Sep | **25 Sep** | **30 Sep** | **27 Sep** |

 Secondly, the fact that the rupture is also synchronous with a very important change in Burgundian vinoculture practices is stricking and suggests a very probable anthropogenic explanation that we have expressed in the paper. As in most of all prestigious vineyards throughout Europe, the style of Beaune wines shifted at that time towards stronger and more long-term keeping wines in comparison with previous centuries, which might eventually led winegrowers to harvest a bit later in order to pick up fully matured grains. It does not exist any document ordering a general delay of the harvest in the beginning of the 18[th] century, as you rightly suggest could be a proof. However, this evolution of the practice has been largely documented in the literature (Dion 1959, Lachiver 1988).

At this point, we think that the historical approach of carefully assessing the raw material is the more reliable methodology.

**4. Statistical modelling**

Thanks for the input. We will be more explicit in the formulation of the model. For instance, it is the natural logarithm that is used in our transformation. Note that the transformations are used for modelling GHD from temperature. Here is a more complete description

Modeling GHD from temperature:

Model A $\quad\quad D = c_0 + c_1 T_{Mar} + c_2 T_{Apr} + c_3 T_{May} + c_4 T_{Jun} + c_5 T_{Jul}$

Model B $\quad\quad D' = c_0 + c_1 T_{Mar}' + c_2 T_{Apr}' + c_3 T_{May}' + c_4 T_{Jun}' + c_5 T_{Jul}'$

Modeling April to July temperature from GHD (note that this is done in T and D, not T' and D', so the manuscript is correct):

Model A $\quad\quad T_{AMJJ} = c_0 + c_1 D$

Model B $\quad\quad T_{AMJJ} = \sum T_{AMJJ,i} \, w_{i,j} / \sum w_{i,j}$

where i are the 7440 preindustrial model years in CCC400 and $w_{i,j}$ is the weight each model year gets for each reconstruction year. We will then rephrase Model B for temperature in a Bayesian formulation, as suggested by the reviewer, and will add this to the description. This is a helpful suggestion.

**References:**

Angot, A.: Études sur les vendanges en France, Annales du bureau central météorologique de France, année 1883, B29-B120, 1885.

Daux, V., Garcia de Cortazar-Atauri, I., Yiou, P., Chuine, I., Garnier, E., Le Roy Ladurie, E., Mestre, O., and Tardaguila, J.: An openaccess database of grape harvest dates for climate research: data description and quality assessment, Clim. Past, 8, 1403–1418, doi:10.5194/cp-8-1403-2012, 2012.

Dion, R. : Histoire de la vigne et du vin en France, des origines au XIX[e] siècle, Paris, chez l'auteur, 1959.

Dufour, M. L.: Problème de la variation du climat. Bulletin de la Société Vaudoise des Sciences naturelles, 10, 359–556, 1870.

Garnier, E., Daux, V., Yiou, P., and Garcia de Cortázar-Atauri, I.: Grapevine harvest dates in Besançon (France) between 1525 and 1847: social outcomes or climatic evidence?, Climatic Change, 104, 703–727, doi:10.1007/s10584-010-9810-0, 2011.

Labbé, T., Gaveau, F.: Les dates de vendange à Dijon : établissement critique et révision archivistique d'une série ancienne, Revue historique, 657, 19-51, 2011.

Lachiver, M. : Vins, vignes et vignerons. Histoire du vignoble français, Paris, Fayard, 1988.

Le Roy Ladurie, E.: Times of Feast, Times of Famine: A History of Climate since the Year 1000, Allen & Unwin, London (Original: Histoire du climat depuis l'an mil, Paris, Flammarion, 1971.

Možny, M., Brázdil, R., Dobrovolný, P., Trnka, M.: April-August temperatures in the Czech Lands, 1499-2015, reconstructed from grape-harvest dates, Climate of the Past, 12/7, 1421-1343, doi:10.5194/cp-12-1421-2016, 2016.

Wetter, O., Pfister, C.: An underestimated record breaking event – why summer 1540 was likely warmer than 2003. Clim Past, 9, 41–56, 2013.

---

## Author Comment (AC3) · 27 May 2019

We first thank the second referee for taking time to read our paper and for his positive encouragements and comments. In the following lines we give some answers to the three main concerns pointed out in the comment.

1. **"It is particularly interesting that the new reconstruction does not corroborate the putative heatwave in 1540 CE […] ; an aspect that should be discussed in a more balanced and nuanced way than currently done."**

Natural proxy data sometimes underrate extreme events that are fully described as such by historical sources. Concerning the extreme temperature and precipitation conditions of the 1540 summer, this controversy has already been discussed in different ways (Wetter et al. 2015; Orth et al. 2016). Wetter et al. 2014 have actually demonstrated from a large array of narrative sources that the spring-summer temperature and precipitation in 1540 have been outstanding and likely more extreme than in 2003 in a large part of Europe, even thought European tree ring width (TRW) series show little evidence of a megadrought in this same year (Büntgen et al. 2015). However, close comparisons of negative TRW extremes showing droughts with documentary data shows important disagreements or inconsistency. TRW may provide responses lagged for one year because tree growth integrate effects from previous year climatic and ecological conditions leading to lagged autocorrelations (Frank et al. 2007; Pfister et al. 2015).

That the heatwave also struck the region of Burgundy in 1540 is not to be discussed. The church of Notre-Dame of Beaune organised 8 processions to call for rains from the beginning of May to the end of August (Labbé, Gaveau 2011). In the near city of Besançon (c. 100 km eastwards) a chronicler wrote that warm temperatures lasted from April to November and that the heatwave was hardly bearable during summer (Wetter et al. 2014). Consequently, it is true that we might expect the grape harvest date in 1540 to be in top three of the earlier GHD of the entire series, when it ranks only 19th. We argue however in section 4.5 of the article that the extreme conditions and hydric stress of 1540 certainly contributed to block the development of the grape in Burgundy which can explain why the harvest did not occur as early as might be expected. In particular, it can be concluded from the sources that mature berries were dried out in early August, after a completely rainless month of July involving temperatures beyond 40°C leading to frequent forest fires. Vine-growers stopped picking grapes until the next rain-spell to get more juice into the berries. Thus, rainfall rather than temperature determined to begin of harvest. As with TRW, it is important to assume that GHD may integrate ecological factors influencing the growth of the vineyards. We believe this is an important improvement in GHD series analysis method that calls for further researches in the future.

We agree however with the reviewer that this question has to be expressed in a more nuanced way in the text of the article and we will add in section 4.5 this sentence: "It is puzzling that the exceptional heat and drought in 1540 ranks only 19th in the statistic. Possible reasons were brought forward by Büntgen et al. (2015), though their arguments are thought to be questionable (Pfister et al. 2015). The modelling of 1540 attempted by Orth et al. (2016) provides a more balanced view. In particular, Wetter et al. (2014) demonstrated from the sources that precipitation, not temperature mattered for the grape harvest in 1540, as many grapes were dried out in early August".

2. **"[…] the article would benefit from an assessment of the persistence (i.e. autocorrelation structure) in Beaune's new GHD-based April-July temperature reconstruction compared to the possibly inflated long-term persistence in tree ring width records"**

This is a helpful suggestion that can improve the analysis of the Beaune GHD-based AMJJA temperature reconstruction. We will add the following paragraph in section 4.3

"The autocorrelation structure of the Beaune based temperature reconstruction (black in attacked figures; regression model A and B lead to equal results) is very similar to the tree-ring reconstruction from the Pyrenees (blue) but has clearly less autocorrelation than the tree-ring reconstruction from the Lötschental, Switzerland (red). The time series reveal the autocorrelation differences in their multi-decadal and lower variability (same colors as above and anomalies with respect to 1961-90). The Lötschental time series experiences much more low frequency variability than the Beaune and Pyrenees records. However, it is hard to argue that one of them should be the correct one. All have uncertainties with regard to their low frequency behaviour. In the tree-ring series many studies discuss these issues that stem from age detrending, temporally inconsistent tree-age distributions, etc. On the other hand documentary data for GHD are also not free from issues concerning low frequency variability. In the case of vine there may be adaptation, breeding, gene (de)activation processes over the decades that may dampen low frequency variability. There may also be changes in taste altering harvest dates in both directions.

Another point worth mentioning in comparison with tree-ring reconstructions is that the Beaune temperature reconstruction rather underestimates interannual variability (SD=0.58 K, period 1659-2007) compared to the Paris observations (SD=0.87 K) whereas both tree-ring reconstructions overestimate it (both SD=1.10 K)."

[Figure]

*Figure 1 : Autocorrelation structure of Beaune GHD-based temperature reconstruction (in black), Löchtental tree-ring-based temperature reconstruction (in red) and Pyrénées tree ring-based temperature reconstruction (in blue)*

3. **"I kindly ask the authors to consider the spatially explicit European summer temperature reconstruction by Luterbacher et al. (2016) for comparison**

In the article we have compared the Beaune record with the two closest proxy records, which actually are the basis for the Luterbacher reconstruction, too (i.e. the Alps and Pyrenees, east and south-west of Beaune respectively). Luterbacher et al. do not have any additional proxy information to north or west. Hence, the interpolation to a coarse grid, even with a sophisticated method, could hardly add significant information at our site.

**References:**

Büntgen, U., Tegel, W., Carrer, M., Krusic, P. J., Hayes, M., Esper, J.: Commentary to Wetter et al. (2014): Limited tree-ring evidence for a 1540 European 'Megadrought', Clim. Chang., 131/2, 183-190, doi: 10.1007/s10584-015-1423-1, 2015.

Labbé, T., Gaveau, F.: Les dates de vendange à Beaune (1371-2010). Analyse et données d'une nouvelle série vendémiologique, Revue historique, 666, 333-367, 2013.

Luterbacher, J., Werner, J.-P., Smerdon, J. E., Fernández-Donado, L., González-Rouco, F. J., Barriopedro, D., Ljungqvist, F.C., Büntgen, U., Zorita, E., Wagner, S., Esper, J., McCarroll, D., Toreti, A., Frank, D., Jungclaus, J. H., Barriendos, M., Bertolin, C., Bothe, O., Brázdil, R., Camuffo, D., Dobrovolný, P., Gagen, M., García-Bustamente, E., Ge, Q., Gómez-Navarro, J. J., Guiot, J., Hao, Z., Hegerl, G. C., Holmgren, K., Klimenko, V. V., Martín-Chivelet, J., Pfister, C., Roberts, N., Schindler, A., Schurer, A., Solomina, O., Von Gunten, L., Wahl, E., Wanner, H., Wetter, O., Xoplaki, E., Yuan, N., Zanchettin, D., Zhang, H., Zerefos, C. : European summer temperatures since Roman times, Environ. Res. Lett., 11/2, 024001, 2016.

Pfister, C., Wetter, O., Brázdil, R., Dobrovolný, P., Glaser, R., Luterbacher, J., Seneviratne, S. I., Zorita, E., Alcoforado, M.-J., Barriendos, M., Bieber, U., Burmeister, K. H., Camenisch, C., Contino, A., Grünewald, U., Herget, J., Himmelsbach, I., Labbé, T., Limanówka, D., Litzenburger, L., Kiss, A., Kotyza, O., Nordli, Ø., Pribyl, K., Restö, D., Riemann, D., Rohr, C., Werner, S., Spring, J.-L., Söderberg J.,

Wagner, S., Werner, J. P.: Tree-rings and people – different views on the 1540 Megadrought. Reply to Büntgen et al. 2015, Clim. Chang., 131/2, 191-198, doi: 10.1007/s10584-015-1429-8, 2015.

Wetter, O., Pfister, C., Werner, J.P., Zorita, E., Wagner, S., Seneviratne, S.I., Herget, J., Grünewald, U., Luterbacher, J., Alcoforado, M.J., Barriendos, M., Bieber, U., Brázdil, R., Burmeister, K.H., Camenisch, C., Contino, A., Dobrovolný, P., Glaser, R., Himmelsbach, I., Kiss, A., Kotyza, O., Labbé, T., Limanówka, D., Litzenburger, L., Nordli, Ø., Pribyl, K., Retsö, D., Riemann, D., Rohr, C., Siegfried, W., Söderberg, J., Spring, J.L.: The year-long unprecedented European heat and drought of 1540 – a worst case, Clim Chang., 125, 349-363, doi:10.1007/s10584-014-1184-2, 2014.

---

## Author Response (AR1)

**Point-by-point reply to Short Comment #1:**

1. Could you later provide a much better map of the study area than the present Fig. 1?

In the final version of the article, we have added a precise map of the Beaune vineyard, with indication of altitudes and relief (Fig. 2)

[Figure]

*Figure 1: The vineyard of the city of Beaune*

**Point-by-point reply to Referee Comment #1**

1. Improved introduction and clarification of notation

See our comment in the Reply
We have improve the introduction with the addition at the very beginning of the introduction of the sentences "Since the Middle Ages the opening day of the grape harvest is each year, in a given territory, the outcome of a collective decision. Either in pre-industrial history, when authorities set each year an official ban after which it was permitted for everybody to pick up the grapes, and in recent history, the opening day of grape harvest has always induced an important amount of documentary data."

2. Consistency of determining the GHD

See our comment in the Reply
We have added this table in the supplementary material (Table S 1), to illustrate our methodology.

*Table S 1 : Comparison between dates of the last meeting of the Notre-Dame of Beaune church chapter before the grape harvest and the official ban harvest set by the city council (1554-1620)*

|  | 1554 | 1555 | 1557 | 1558 | 1569 | 1574 | 1583 | 1613 | 1617 | 1619 | 1620 |
|---|---|---|---|---|---|---|---|---|---|---|---|
| Ban dates set by the city council of Beaune | 3 Sep | 19 Sep | 7 Sep | 6 Sep | 9 Sep | 13 Sep | 5 Sep | 21 Sep | 27 Sep | 20 Sep | 25 Sep |
| Dates given by the protocols of the church of Notre-Dame | 23 Aug | 13 Sep | 3 Sep | 2 Sep | 2 Sep | 15 Sep | 1 Sep | 18 Sep | 20 Sep | 6 Sep | 16 Sep |
| Estimated dates | 26 Aug | 16 Sep | 6 Sep | 5 Sep | 5 Sep | 18 Sep | 8 Sep | 26 Sep | 28 Sep | 14 Sep | 24 Sep |

3. Harmonisation of time series

See our comment in the Reply
To illustrate the special behaviour of the Beaune GHD time series in comparison with other European GHD times series, we have improved the Table 1 in the final version of the article. The new Table 1 contains a comparison with the Aubonne series, and a new column "Difference of days between Period 1 and Period 2", which did not appear in the discussion paper.

*Table 2: Comparison of mean GHD time series in Beaune (this article), Aubonne (Angot 1885), Salins (Angot 1885), Switzerland (Wetter et al. 2013) and Czech Lands (Mózny et al. 2016).*

| | Period 1: 1599-1717 | Period 2: 1718-1875 | 1354-1717 | 1718-2018 | Difference of days between period 1 and period 2 |
|---|---|---|---|---|---|
| Mean Beaune GHD | 21 Sep | 28 Sep | 20 Sep | 27 Sep | +7 |
| Mean Swiss GHD | 20 Oct | 15 Oct | | | -5 |
| Mean Czech Lands GHD | 14 Oct | 14 Oct | | | 0 |
| Mean Salins GHD | 11 Oct | 09 Oct | | | -2 |
| Mean Aubonne GHD | 21 Oct | 18 Oct | | | -3 |

Furthermore, we have partially rephrased paragraph 2.6, explaining the reasons why we have chosen to homogenised the Beaune GHD raw data adding 7 days to all dates prior to 1718.

l. 189-192: "Otherwise it would have mean April-to-August GHD temperature based reconstructions with maxima and minima c. 1°C warmer before 1718 than afterwards on the decadal scale, even though temperatures measurements available in France since 1658 do not provide any evidences of such a warming during the period 1658-1718."

l. 202-205: "In the course of the 18th century, agronomists all underlined the fact that more mature grapes favour the process of fermentation and can prevent the acidification of wines. The works of Jean-Antoine Chaptal (Chaptal 1807) who gave his name to the process of *chaptalisation*, i.e. addition of sugar in grape must, illustrate this evolution."

**4. Statistical modelling**

See our comment in the Reply

Section 3.2 has been largely rephrase accordingly to the reviewer comment. Additionally we have reworked the terminology of Model A and Model B that was found too complicated. (especially as the two denote something different in the forward and backward way). Therefore we have chosen a new and more intuitive terminology:

$F_{linear}$, $F_{transformed}$, $F_{transformed.EKF400}$, $F_{transformed.CCC400}$, $B_{linear}$, $B_{weighted.simulations}$

Here F stands for forward and B for backward, with ".CCC400 and .EKF400" we denote the application of the calibrated model to these data sets. The rest (linear, transformed, weighted simulations) should be self-explanatory. Furthermore, we now give the explicit regression equation with the fitted coefficients for all models.

**Point-by-point reply to Referee Comment #2**

1. "It is particularly interesting that the new reconstruction does not corroborate the putative heatwave in 1540 CE [...] ; an aspect that should be discussed in a more balanced and nuanced way than currently done."

See our comment in the reply.
We have given more information in the final version of the article on this especially important issue.
We have rephrased Section 4.5 with the following input:

[revised manuscript text omitted]

2. "[…] the article would benefit from an assessment of the persistence (i.e. autocorrelation structure) in Beaune's new GHD-based April-July temperature reconstruction compared to the possibly inflated long-term persistence in tree ring width records"

See our comment in the Reply
We entirely agree with the proposition made by the reviewer. Therefore, we have added in Section 4.3 the following paragraph that develops some considerations on this issue:

"The autocorrelation structure of the Beaune GHD-based temperature reconstruction (black in figure S2; regression model F and B lead to equal results) is very similar to the tree-ring reconstruction from the Pyrenees (blue) but has clearly less autocorrelation than the tree-ring reconstruction from the Lötschental, Switzerland (red). The Lötschental time series experiences much more low frequency variability than the Beaune and Pyrenees records. However, it is hard to argue that one of them should be the correct one. All have uncertainties with regard to their low frequency behaviour. In the tree-ring series many studies discuss these issues that stem from age detrending, temporally inconsistent tree-age distributions, etc. On the other hand documentary data for GHD are also not free from issues concerning low frequency variability. In the case of vine there may be adaptation, breeding, gene (de)activation processes over the decades that may dampen low frequency variability. There may also be changes in taste altering harvest dates in both directions.

Another point worth mentioning in comparison with tree-ring reconstructions is that the Beaune temperature reconstruction rather underestimates interannual variability (SD=0.58 K, period 1659-2007) compared to the Paris observations (SD=0.87 K) whereas both tree-ring reconstructions overestimate it (both SD=1.10 K)."
This paragraph is accompanied with a new figure provided as supplementary material (Figure S 2; see below).

[Figure]

*Figure S 2 : Autocorrelation structure of Beaune GHD-based temperature reconstruction (in black), Löchtental tree-ring-based temperature reconstruction (in red) and Pyrénées tree ring-based temperature reconstruction (in blue)*

3. "I kindly ask the authors to consider the spatially explicit European summer temperature reconstruction by Luterbacher et al. (2016) for comparison

See our comment in the reply: "In the article we have compared the Beaune record with the two closest proxy records, which actually are the basis for the Luterbacher reconstruction, too (i.e. the Alps and Pyrenees, east and south-west of Beaune respectively). Luterbacher et al. do not have any additional proxy information to north or west. Hence, the interpolation to a coarse grid, even with a sophisticated method, could hardly add significant information at our site."
Accordingly we did not change anything on this point.

[revised manuscript text omitted]